# Debiasing Pretrained Generative Models by Uniformly Sampling Semantic Attributes

Walter Gerych[1] [*], Kevin Hickey[1], Luke Buquicchio[1], Kavin Chandrasekaran[1], Abdulaziz Alajaji[2],
Elke Rundensteiner[1], Emmanuel Agu[1]

[1]Worcester Polytechnic Institute, Worcester, MA
[2]King Saud University, Riyadh, Saudi Arabia

## Abstract

Generative models are being increasingly used in science and industry applications. Unfortunately, they often perpetuate the biases present in their training sets, such as societal biases causing certain groups to be underrepresented in the data. For instance, image generators may overwhelmingly produce images of white people due to few non-white samples in their training data. It is imperative to debias generative models so they synthesize an equal number of instances for each group, while not requiring retraining of the model to avoid prohibitive expense. We thus propose a *distribution mapping module* that produces samples from a *fair noise distribution*, such that the pretrained generative model produces *semantically uniform* outputs - an equal number of instances for each group - when conditioned on these samples. This does *not* involve retraining the generator, nor does it require *any* real training data. Experiments on debiasing generators trained on popular real-world datasets show that our method outperforms state-of-the-art approaches.

## 1 Introduction

**Background.** Generative models have become a cornerstone of modern machine learning, allowing for the synthesis of realistic data for many domains, including images [19, 35], audio [22], and text [11, 4]. However, even leading generative models often reproduce the biases present in their training data [13, 26], such as image generation models strongly over-representing white males [24]. As generative models are increasingly used for data augmentation to train downstream models [5] in domains from scientific to medical fields [25, 9], biased synthesized data could lead to results that are skewed or inaccurate. This can exacerbate existing issues such as facial recognition models performing significantly worse on non-white individuals [29] or healthcare models being much less accurate for certain minority groups [23]. Further, with the rapid growth of generative models in commercial applications, the potential financial, legal and ethical costs of biased outputs are significant. Thus, it is imperative to develop methods that mitigate bias in generative models to ensure that their outputs are fair and equitable by generating a roughly equal number of samples of each group. We call such outputs *semantically uniform*, and the attribute that they are uniform over - such as gender or race - the *semantic attribute*.

**State-of-the-art.** Existing methods for addressing bias in generative models often train a new model from scratch [27, 37, 2], though this is computationally expensive, requires significant labeled data, and wastes resources already previously spent training the existing (biased) generative model. Latent attribute editing methods modify the samples in the latent space of generative models to produce controlled changes in the output, and could potentially be used to correct for bias [17]. However, this requires making limiting assumptions such as that semantic attributes correspond to *linear* directions

---

[*]Corresponding author: wgerych@wpi.edu

37th Conference on Neural Information Processing Systems (NeurIPS 2023).

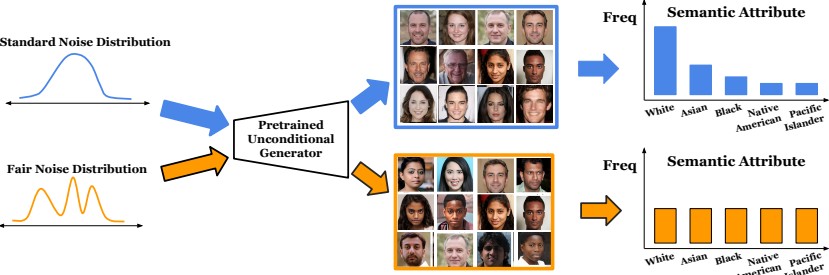

Figure 1: A generator conditioned on a fair noise distribution yields outputs that are uniform over a semantic attribute (i.e. race).

in the latent space [17, 38, 6, 33, 39, 8]. While some recent advances have been made in uniformly sampling the output space of a pretrained model [15, 16], they can fail to yield uniform samples over *semantic attributes* [15]. For example, they will make an image generator's output uniform over a manifold in the pixel space, but may still overproduce images of white individuals as the output won't be uniform over the *semantic attribute of race*.

**Problem statement.** Recently, numerous pretrained generative models and classifiers have been released for a variety of domains. For instance, there are publicly available generators for images of faces [19] followed by classifiers that predict attributes such as race and age of faces [36]. Here, we assume access to a pretrained generative model $G_\theta$ that maps a noise space $\mathcal{Z}$ to a feature space $\mathcal{X}$, and a pretrained classifier $C_\phi$ that maps $\mathcal{X}$ to a discrete semantic attribute space $\mathcal{Y}$. Additionally, as error rates are often reported for released models, we assume that the class-conditional errors; i.e., the probability that an instance actually belongs to group $j$ given that the classifier incorrectly predicted group $i$, is known. Our goal is to model a distribution $\mathbb{Q}$ such that for $\mathbf{z} \sim \mathbb{Q}$, $G_\theta(\mathbf{z})$ produces samples that are uniform over the attribute space $\mathcal{Y}$, using only predictions from $C_\phi$ to guide the output of $G_\theta$. For instance, if there are five discrete semantic groups in $\mathcal{Y}$, $G_\theta$ will produce an equal number of instances belonging to each group when conditioned on draws from $\mathbb{Q}$. We call $\mathbb{Q}$ a *fair noise distribution*; see Figure 1 for an example.

**Challenges.** Our task of producing samples that are semantically uniform has three major hurdles:

1. *Expensive retraining:* It is often prohibitively expensive to retrain large generative models in terms of computational resources and time.
2. *Inaccessible training data:* The data used to train a released generator is typically unavailable; either due to being proprietary or simply too large of a volume for most practitioners to utilize. Thus, we do not have adequate data available to tune the generator or classifier.
3. *Inaccurate classifier:* As we do not have any samples of real data available, we must rely on the possibly imperfect semantic attribute classifier $C_\phi$ to provide labels. However, since classifiers may produce inaccurate predictions - especially on underrepresented groups of $\mathcal{Y}$, this can cause incorrect estimations of the number of instances for each group.

**Proposed approach.** We propose to train a distribution mapper network $M_\omega : \mathcal{Z} \rightarrow \mathcal{Z}$ that transforms draws from a standard noise distribution into draws from a fair noise distribution, such that once noise samples are transformed by $M_\omega$ they condition the generator to produce a roughly equal number of instances for each semantic group in $\mathcal{Y}$. We achieve this by first constructing a dataset of instances from $\mathcal{Z}$ that follow a fair noise distribution, which allows us to estimate the true distribution of the semantic attribute given an imperfect classifier. Using our proposed strategy, noise samples are collected such that the pretrained classifier's (corrected) distribution is uniform. Then, we use this fair noise dataset to train $M_\omega$ to inexpensively sample new instances from the fair noise distribution. The output of $M_\omega$ serves as input to the pretrained generator, which yields samples that are uniform over the space of the semantic attribute. Importantly, this approach does not require retraining the main generative model, thus incurring minimal costs. Our method works without requiring any real training data. Relying only on a pretrained generator and classifier pair, we circumvent the need to acquire any real data (labeled or otherwise) even using only an imperfect classifier.

**Contributions.** In this work, we:

- Show how to construct a fair noise dataset that produces semantically uniform synthetic outputs when passed through a generator, assuming an accurate classifier is available.

- Design a method to correct for the error incurred by an imperfect classifier, to construct a dataset of samples that come from a fair noise distribution.

- Propose an approach for training a distribution mapping network to sample from a fair noise distribution on-the-fly.

- Demonstrate the utility of our approach on a range of real-world datasets and released generators.

## 2 Problem Definition

Assume we are given a pretrained generative model $G_\theta$ that maps a latent space $\mathcal{Z}$ to a feature space $\mathcal{X}$. In general, we assume that $\mathcal{Z}$ is of a lower dimensionality than $\mathcal{X}$. Additionally, we assume we are given a pretrained classifier $C_\phi$ that maps $\mathcal{X}$ to a semantic attribute space $\mathcal{Y}$, where $\mathcal{Y}$ is the set of *group* (*class*) labels. Thus, $\mathcal{Y} = \{Y_1, Y_2, \ldots, Y_N\}$, where $Y_i$ is the label of the $i$th group. When not otherwise ambiguous, we will refer to group $Y_i$ as group $i$. Let $\mathbf{y}$ and $\hat{\mathbf{y}}$ be the random variables indicating the true semantic attribute label and the predicted label from $C_\phi$ respectively. Let $E$ be the prediction-conditional error rates of $C_\phi$, with $E$ being a left stochastic matrix such that $E_{i,j} = P(\mathbf{y} = i | \hat{\mathbf{y}} = j)$. Let $C_\phi$ be a better-than-random classifier, i.e., for $N$ groups, $P(\mathbf{y} = i | \hat{\mathbf{y}} = i) > \frac{1}{N}$ for all groups $i$. Thus, $E$ is by definition a diagonally dominant matrix. As it is typical for developers to report the error rates of their classifiers, we assume that $E$ is known. While these error rates are typically reported for the classifier's training distribution, we can correct $E$ to apply to the generated distribution under a label shift assumption [40]. See the Appendix for details on this and for the proofs of the upcoming theorems. Lastly, let $C'$ denote an ideal *perfect* classifier that always correctly predicts $\mathbf{y}$ with zero error. $C'$ is hypothetical and unavailable to us.

Our goal here is to sample from a Fair Noise Distribution, defined as follows:

**Definition 1** (Fair Noise Distribution). *For a generative model $G_\theta : \mathcal{Z} \to \mathcal{X}$, a distribution $\mathbb{Q}$ over $\mathcal{Z}$ is a **Fair Noise Distribution** with respect to $\mathcal{Y}$ if for $\mathbf{z} \sim \mathbb{Q}$, $C'(G_\theta(\mathbf{z})) \sim \mathrm{Unif}(\mathcal{Y})$, where $\mathrm{Unif}(\mathcal{Y})$ is the uniform distribution over the groups in $\mathcal{Y}$.*

Intuitively, a distribution $\mathbb{Q}$ is a Fair Noise Distribution if $G_\theta$ produces samples that are uniform over the semantic space $\mathcal{Y}$ when conditioned on $\mathbb{Q}$. For example, if $G_\theta$ synthesizes images of people and $\mathcal{Y}$ is the set of races, then $\mathbb{Q}$ is a Fair Noise Distribution if conditioning on it makes $G_\theta$ produce a roughly equal number of images of people from each race.

More realistically, we want to find a $\mathbb{Q}$ that is easy to sample from and produces reasonable variance from the generator. That is, it does not yield only one unique example of each group, which would otherwise make the problem trivial. Notably, we assume that we do *not* have any real training data (i.e., no real samples from $\mathcal{X}$). Also, we do not aim to retrain $G_\theta$, i.e., not change the parameters $\theta$ of $G_\theta$. Additionally, we do not make any assumptions on the differentiability of $G_\theta$ or $C_\phi$.

## 3 Methodology

Our approach for sampling from a Fair Noise Distribution $\mathbb{Q}$ hinges on training a distribution mapping function $M_\omega$ such that $M_\omega(\mathbf{z}) \sim f\mathbb{Q}$, where $\mathbf{z}$ is a draw from the original conditioning distribution of $G_\theta$. The functional form of $M_\omega$ has many options; for instance, $M_\omega$ can be a GAN generator [12], a VAE [21], a DDPM [14], or a normalizing flow model [34, 7]. No matter which form is chosen, we will need a dataset of samples drawn from $\mathbb{Q}$ to train $M_\omega$.

### 3.1 Collecting Fair Samples Using Imperfect Classifiers

A naive method for collecting a dataset of samples distributed according to a Fair Noise Distribution $\mathbb{Q}$ is given in Algorithm 1. The basic approach is to continuously sample $\mathbf{z}$ from the noise distribution, collect the generator's output for each draw, and use the classifier to determine the value of the semantic attribute corresponding to each noise draw. As a result, samples of $\mathbf{z}$ corresponding to each group are saved to dataset $\mathcal{D}_\mathbb{Q}$ until $\mathcal{D}_\mathbb{Q}$ has $S$ number of samples for each $y \in \mathcal{Y}$. This procedure results in samples of noise that once passed through $G_\theta$ will be uniform across the semantic attribute

---

**Algorithm 1** Naively collect data from $\mathbb{Q}$

---

1: **procedure** NAIVE_Q_DATASET($G_\theta, C_\phi, S$)               ▷ Colect from $\mathbb{Q}$ by trusting $C_\phi$
2:    **Input:** Pretrained generator $G_\theta$ and classifier $C_\phi$, number of samples $S$ for each attribute
3:    **Output:** Dataset with $S$ samples for each $y \in \mathcal{Y}$
4:    $Q\_dataset \leftarrow dict()$                      ▷ Initialize empty dictionary for the dataset
5:    **for** $group \in \mathcal{Y}$ **do**
6:       $Q\_dataset[group] = [\,]$                      ▷ Initialize every group as empty array
7:    **end for**
8:    **while** $\exists\, key \in Q\_dataset.keys$ s.t. $length(Q\_dataset[key]) < S$ **do**
9:       $z \sim \mathbb{P}_{\text{noise}}$                                 ▷ Sample noise
10:       $group = C_\phi(G_\theta(z))$                        ▷ Get the predicted group of $G_\theta(z)$
11:       **if** $length(Q\_dataset[class]) < S$ **then**
12:          $Q\_dataset[class].append(z)$     ▷ If $< S$ samples for that class, add $z$ to the dataset
13:       **end if**
14:    **end while**
15:    **return** $Q\_dataset$                      ▷ Return dataset with $S$ samples for each group in $\mathcal{Y}$
16: **end procedure**

---

space *according to* $C_\phi$; i.e., for $\mathbf{z} \sim \mathcal{D}_\mathbb{Q}$, $C_\phi(G_\theta(\mathbf{z})) \sim \text{Unif}(\mathcal{Y})$. However, this will only yield samples from a Fair Noise Distribution in the case where $C_\phi$ is a perfect classifier (such that $E$ is the identity matrix). If $C_\phi$ is an imperfect classifier then the distribution may not be truly fair. For instance, if $C_\phi$ often incorrectly predicts group $j$ as group $i$, then instances with attributes matching group $i$ may be more prevalent than those for group $j$.

Fortunately, we can utilize knowledge from the prediction-conditional error rates $E$ to sample a dataset of noise that will yield more semantically uniform generated instances despite the noisy predictions of $C_\phi$. To achieve this, we use a weighted sample of datapoints predicted for each group. Let $\mathbb{P}_{z|C_\phi=i}$ be a distribution over $\mathcal{Z}$ such that the imperfect classifier's prediction of generated samples arising from this distribution are all of group $i$; $C_\phi(G_\theta(\mathbf{z})) = i$ for $\mathbf{z} \sim \mathbb{P}_{z|C_\phi=i}$. In addition, let $\mathbb{Q}^\lambda = \sum_{i=1}^{|\mathcal{Y}|} \lambda_i \mathbb{P}_{z|C_\phi=i}$, such that $\lambda_i > 0\ \forall\ i$ and $\sum_{i=1}^{|\mathcal{Y}|} \lambda_i = 1$.

We can in many instances find values of $\lambda = \{\lambda_1, \lambda_2, \ldots, \lambda_{|\mathcal{Y}|}\}$ such that $\mathbb{Q}^\lambda$ is a Fair Noise Distribution, as stated in Lemma 1:

**Lemma 1.** *Let $\mathbf{1}^{|\mathcal{Y}|}$ be the vector of length $|\mathcal{Y}|$ such that every element is 1. Let $\text{cone}(E) = \{\sum a_i E_{:,i}|a_i \in \mathbb{R}_{\geq 0}\}$ be the finite convex cone generated by the columns of the prediction-conditional error matrix. If $\mathbf{1}^{|\mathcal{Y}|} \in \text{cone}(E)$, then $\exists\ \lambda$ such that $\mathbb{Q}^\lambda$ is a Fair Noise Distribution.*

Additionally, we can show that in the case where $\mathbf{y}$ is binary and $C_\phi$ is better-than-random, we can *always* find $\lambda$ such that samples drawn from $\mathbb{Q}^\lambda$ will definitely yield generated samples that are uniform over the semantic attributes, as shown by the following lemma.

**Lemma 2.** *Let $|\mathcal{Y}| = 2$ and let $C_\phi$ be better-than-random, such that the diagonal elements of $E$ are each greater than $\frac{1}{2}$. Then, $\exists\ \lambda$ such that $\mathbb{Q}^\lambda$ is a Fair Noise Distribution.*

The proof of Lemma 2 comes from showing that $\langle 1, 1 \rangle$ is always in the finite convex cone generated by $C_\phi$'s prediction-conditional error matrix, with Lemma 1 implying that this property guarantees that $\mathbb{Q}^\lambda$ will yield samples that produce semantically uniform generated instances.

Even in the cases where we cannot guarantee that there exists an ideal $\lambda$, we can still find values for $\lambda$ that will yield a distribution that is *as close as possible* to a uniform distribution over the semantic space, with the difference from uniformity measured by KL divergence. For this, let us define a Minimally-Unfair Noise Distribution.

**Definition 2** (Minimally-Unfair Noise Distribution)**.** *For a generative model $G_\theta : \mathcal{Z} \to \mathcal{X}$, a distribution $\mathbb{Q}^\lambda = \sum_{i=1}^{|\mathcal{Y}|} \lambda_i \mathbb{P}_{z|C_\phi=i}$, $\lambda = \{\lambda_1, \lambda_2, \ldots, \lambda_{|\mathcal{Y}|}\}$, $\lambda_i \in \mathbb{R}_{\geq 0}$, $\sum_{i=1}^{|\mathcal{Y}|} \lambda_i = 1$, is a **Minimally-Unfair Noise Distribution** over $\mathcal{Z}$ with respect to $\mathcal{Y}$ if for $\mathbf{z} \sim \mathbb{Q}^\lambda$, $\lambda = \underset{\lambda}{\arg\min}\ \text{KL}\{C'(G_\theta(\mathbf{z}))||\text{Unif}(\mathcal{Y})\}$.*

---

**Algorithm 2** Collecting a dataset of samples from $\mathbb{Q}^\lambda$

---

1: **procedure** WEIGHTED_Q_DATASET($\{\mathcal{D}_{z|C_\phi=i}\}_{i=1}^{|\mathcal{Y}|}, S$)
2:     **Input:** $\mathcal{D}_{z|C_\phi=i}$ for $i$ from $1 \to |\mathcal{Y}|$: Dataset of noise samples that produce generated
    instances which $C_\phi$ classifies as group $i$; $S$: number of instances we wish to sample from $\mathbb{Q}^\lambda$
3:     **Output:** Dataset with $S$ samples from $\mathbb{Q}^\lambda$
4:     $\lambda \leftarrow argmax H(\mathbb{P}_E^\lambda)$
5:     $Q\_dataset = []$                                   $\triangleright$ Initialize the $\mathbb{Q}^\lambda$ as an empty array
6:     **for** $m \leftarrow 1$ to $S$ **do**
7:         $r \leftarrow random\_int(\lambda)$     $\triangleright$ get random int (1 to $|\mathcal{Y}|$) with probability proportional to $\lambda$
8:         $z \sim \mathcal{D}_{z|C_\phi=r}$
9:         $Q\_dataset.append(z)$
10:    **end for**
11:    **return** $Q\_dataset$                          $\triangleright$ Return dataset with $S$ samples from $\mathbb{Q}^\lambda$
12: **end procedure**

---

Intuitively, $\mathbb{Q}^\lambda$ is a Minimally-Unfair Noise Distribution if the values of $\lambda$ yield generated samples with minimal divergence from a semantically uniform distribution, under the constraint that $\mathbb{Q}^\lambda$ is a convex combination of the $\mathbb{P}_{z|C_\phi=i}$'s. This constraint is required so that we can sample from $\mathbb{Q}^\lambda$ using a weighted sampling technique based off of our imperfect classifier $C_\phi$.

Fortunately, we can easily find the values of $\lambda$ that will yield a Minimally-Unfair Noise Distribution. Before deriving the procedure for this, let us define the distribution $\mathbb{P}_E^\lambda$ as the normalized weighted sum of the columns of $E$, where each column $i$ is weighted according to a corresponding $\lambda_i$:

**Definition 3** ($\mathbb{P}_E^\lambda$). *Define $\mathbb{P}_E^\lambda = \sum_{i=1}^{|\mathcal{Y}|} \lambda_i E_{:,i}$ as a distribution over $\mathcal{Y}$ determined by prediction-conditional error matrix $E$ and $\lambda = \{\lambda_1, \lambda_2, \ldots, \lambda_{|\mathcal{Y}|}\}$, $\lambda_i \in \mathbb{R}_{\geq 0}$, $\sum_{i=1}^{|\mathcal{Y}|} \lambda_i = 1$.*

Next, we note that finding the values for $\lambda$ that maximize the entropy of $\mathbb{P}_E^\lambda$ is equivalent to finding the $\lambda$ for which the divergence between $\mathbb{P}_E^\lambda$ and $\mathrm{Unif}(\mathcal{Y})$ is minimized. This is stated formally in the following Proposition.

**Proposition 1.** *If $\lambda^* = \underset{\lambda}{\operatorname{argmax}} H(\mathbb{P}_E^\lambda)$ where $H$ is entropy, then for the same $\lambda^*$ it is true that $\lambda^* = \underset{\lambda}{\operatorname{argmin}} \mathrm{KL}\{\mathbb{P}_E^\lambda || \mathrm{Unif}(\mathcal{Y})\}$.*

Before stating the theorem that directly implies a strategy for learning $\lambda$ for which $\mathbb{Q}^\lambda$ is Minimally-Unfair, all that is left to do is to link $\mathbb{P}_E^\lambda$ with the distribution of $C'(G_\theta(\mathbf{z}))$:

**Proposition 2.** *If $\mathbf{z} \sim \mathbb{Q}^\lambda$, then it is true that $C'(G_\theta(\mathbf{z})) \sim \mathbb{P}_E^\lambda$.*

Proposition 2 states that for a given $\lambda$ the distribution of the perfect classifier $C'$ given samples of the generator conditioned on draws from $\mathbb{Q}^\lambda$ will be distributed according to $\mathbb{P}_E^\lambda$. Now, we state Theorem 1 which directly implies our sampling strategy.

**Theorem 1.** *If $\lambda^* = \underset{\lambda}{\operatorname{argmax}} H(\mathbb{P}_E^\lambda)$, then $\mathbb{Q}^{\lambda*}$ is a Minimally-Unfair Noise Distribution. When $C_\phi = C'$ or $\mathbf{1}^{|\mathcal{Y}|} \in \mathrm{cone}(E)$, then $\mathbb{Q}^{\lambda*}$ is also a Fair Noise Distribution.*

Theorem 1 follows from the preceding Propositions. To see how Theorem 1 implies a strategy for sampling from a Minimally-Unfair Noise Distribution, recall that $\mathbb{Q}^\lambda$ is defined as the mixture of distributions of each $\mathbb{P}_{z|C_\phi=i}$ for $i$ from $1 \to |\mathcal{Y}|$ with $\lambda_i$ as the $i$th mixture weight, where $\mathbb{P}_{z|C_\phi=i}$ is the distribution of $\mathbf{z}$'s that yield generated samples that the noisy classifier predicts as group $i$. The above theorem states that the mixture of these distributions with mixture weights $\lambda$ that produce maximum entropy for $\mathbb{P}_E^\lambda$ will yield a mixture distribution $\mathbb{Q}^\lambda$ that is worse-case Minimally-Unfair and, when possible, will be a perfect Fair Noise Distribution. Thus, to sample from $\mathbb{Q}^\lambda$ all that is needed is to 1) construct datasets $\mathcal{D}_{z|C_\phi=i}$ for $i$ from $1 \to |\mathcal{Y}|$ such that if $\mathbf{z} \in \mathcal{D}_{z|C_\phi=i}$ then $C_\phi(G_\theta(\mathbf{z})) = i$ (i.e., these datasets serve as proxies for $\mathbb{P}_{z|C_\phi=i}$); 2) find $\lambda$ such that $H(\mathbb{P}_E^\lambda)$ is maximized; and 3) perform a weighted sampling from each $\mathcal{D}_{z|C_\phi=i}$ proportional to its corresponding $\lambda_i$. Note that the datasets $\mathcal{D}_{z|C_\phi=i}$ can be formed by an approach similar to

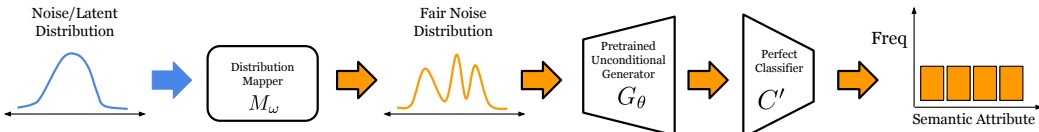

Figure 2: Distribution Mapper $M_\omega$ turns a standard noise distribution into a Fair Noise Distribution.

Algorithm 1. The algorithm returns a dictionary with keys being the elements of $\mathcal{Y}$, and the values of each key returned from that procedure form the corresponding required datasets (i.e., $\mathcal{D}_{z|C_\phi=i}$). A pseudo-code implementation of our approach for constructing a dataset of noise samples from a Minimally-Unfair Noise Distribution $\mathbb{Q}^\lambda$ is given in Algorithm 2.

## 3.2 Training the Distribution Mapper

Let $\mathcal{D}_{\mathbb{Q}^\lambda}$ be a dataset of noise samples from the space $\mathcal{Z}$ distributed according to $\mathbb{Q}^\lambda$, such that $\mathcal{D}_{\mathbb{Q}^\lambda}$ is obtained as described in the previous subsection (i.e., $\mathcal{D}_{\mathbb{Q}^\lambda}$ comes from Algorithm 1 in the case where the error rates $E$ are unknown or $C_\phi$ can be assumed to be an ideal classifier, or from Algorithm 2 otherwise). Now, we train a *distribution mapper* $M_\omega : \mathcal{Z} \to \mathcal{Z}$ such that if $\mathbf{v} \sim \mathbb{P}_{\text{noise}}$ then $M_\omega(\mathbf{v}) \sim \mathbb{Q}^\lambda$, where $\mathbb{P}_{\text{noise}}$ is an easy-to-sample-from noise distribution such as a multivariate Gaussian, or whatever noise distribution was used as input when training $G_\theta$. As previously stated, the functional form of $M_\omega$ and the procedure used to train it is flexible. There is a wide class of generative models and training strategies that could be employed [3]. We chose to model $M_\omega$ as a GAN generator and find the parameters of $M_\omega$ using adversarial training. Thus, we train $M_\omega$ as such:

$$\max_\rho \min_\omega L(M_\omega, F_\rho) = \mathbb{E}_{\mathbf{z} \sim \mathcal{D}_{\mathbb{Q}^\lambda}} \Big[ \log \big( F_\rho(\mathbf{z}) \big) \Big] - \mathbb{E}_{\mathbf{v} \sim \mathbb{P}_{\text{noise}}} \Big[ \log \big( 1 - F_\rho(M_\omega(\mathbf{v})) \big) \Big],$$

where $F_\rho$ is a discriminator network.

Note that $\mathbf{z}$ is a multivariate vector, where the ordering of the elements of the vector are arbitrary (i.e., there is likely no spacial relationship in $\mathbf{z}$). Thus, the inductive bias of convolutional filters used in many leading GANs [20, 19, 32] is not appropriate in this case. Rather, we suggest to train the distribution mapper using a GAN architecture designed for tabular data; notably, CTGAN which has shown to perform well on data where a sequential or spacial inductive bias is not appropriate [43].

After training $M_\omega$ we can sample an arbitrary amount of samples from $\mathbb{Q}$ without applying Algorithm 1 or Algorithm 2, as $M_\omega(\mathbf{v}) \sim \mathbb{Q}^\lambda$ for $\mathbf{v} \sim \mathbb{P}_{\text{noise}}$. Importantly, $C'(G_\theta(M_\omega(\mathbf{v}))) \sim \text{Unif}(\mathcal{Y})$ when $\mathbb{Q}^\lambda$ is a Fair Noise Distribution, and will otherwise have minimum divergence from $\text{Unif}(\mathcal{Y})$ under the constraints given in Definition 2. Notably, we achieve this with *no samples of real data, an imperfect classifier, and without fine-tuning $G_\theta$*. Instead, the only training needed is for $M_\omega$, which in general should require much less complexity (and thus much less cost) than $G_\theta$. Figure 2 shows the Distribution Mapper paired with the pretrained generator $G_\theta$.

## 4 Experiments

We now experimentally evaluate our approach. Details such as hyperparameter choice and architectures are available in the appendix. Code for our method is available in the supplemental material.

We compare our approach on a range of generators: VAE [21], DCGAN [32], Progressive GAN [19], and a Latent Diffusion Model (LDM) [35].

### 4.1 Compared Methods

*Latent Editing (2019).* We apply the commonly used latent editing [17, 38, 6, 33, 39, 8] approach to the task of constructing a dataset with an equal number of instances from each group in $\mathcal{Y}$. While some work uses non-linear directions when editing [1], using linear directions has been shown to work as well in practice [17, 8]. Thus, for each group $y \in \mathcal{Y}$, we fit a linear classifier $K_y$ on noise samples from $\mathbb{P}_{noise}$, with the goal of separating noise instances that are mapped to group $y$ from those mapped to all other groups. After training these classifiers, we can sample a noise instance belonging to a group with the following procedure: First, sample $z \sim \mathbb{P}_{noise}$; if $K_y(z) = y$ then

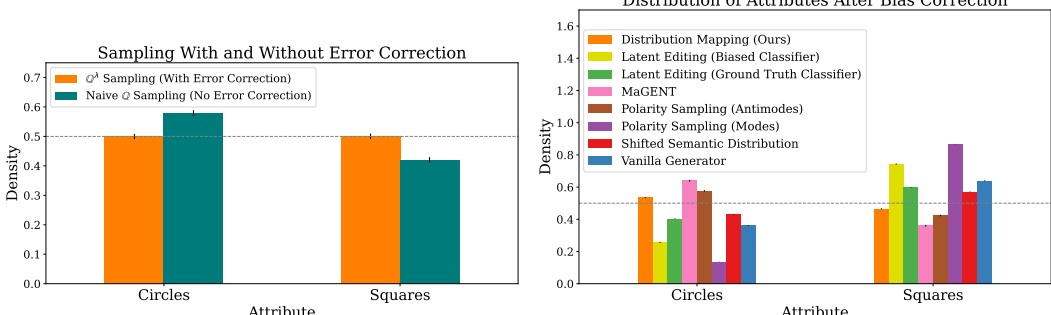

(a) Effect of bias correction; 0.5 (dashed line) is ideal. (b) Comparison of methods; 0.5 (dashed line) is ideal.

Figure 3: a) Effects of applying our bias correction when constructing Fair Noise dataset; b) comparison of methods on the Shapes dataset.

return $z$, otherwise perform latent editing on $z$ until $K_y(z_{new}) = y$. The editing approach is given by $z_{new} = z_{previous} + \epsilon \cdot \eta_y$, where $\eta_y$ is the normal vector to $K_y$'s decision boundary and $\epsilon$ is a step size. We set $\epsilon = 0.1$ in our experiments.

*MaGNET (2022).* This method was recently proposed to uniformly sample the manifold of pretrained generative models [15], using a sampling strategy that leverages the Jacobian of the generative model. The authors argue that while this does not guarantee an equal distribution for each group, *MaGNET* should increase the frequency of disadvantaged groups by sampling more often from low-density regions of the generative manifold.

*Polarity Sampling (2022).* An extension of *MaGNET*, *Polarity Sampling* allows for more controlled sampling over the generative manifold [16]. Sampling is controlled by a parameter $\rho$ where as $\rho$ goes to $-\infty$ modes are sampled from increasingly often, antimodes are sampled from more as $\rho \to \infty$, and the original generative distribution is sampled from for $\rho = 0$. We thus compare against two *Polarity Sampling* settings; *Polarity Mode Sampling* where $\rho = -2$, and *Polarity Antimode Sampling* where $\rho = 2$. By a similar argument used for *MaGNET*, sampling from antimodes may result in minority groups being more represented.

*Shifted Semantic Distribution (2021).* This method provides a training-free approach for debiasing pretrained generative models [42]. The key idea is to fit Gaussian mixture models on the latent space, where these mixtures are fit on regions that correspond to unique values of the semantic attribute. However, unlike our approach, Shifted Semantic Distribution assumes that each attribute class can be uniquely identified by a hyperplane in the noise space.

*Standard Generator.* We compare against unmodified pretrained generators used as initially intended. This is a baseline which other methods should outperform.

## 4.2 Uniformly Sampling From Shapes Dataset

We first evaluate our approach on a dataset of synthetic images. Each image is of either a circle or a square, where the shape has a random size, color, and position in the image. We use the object's shape as the semantic attribute; thus, $\mathcal{Y} = \{\text{'Circle'}, \text{'Square'}\}$. This dataset was first used by Jing et al. [18]. We utilize the VAE Jing et al. compared against in the same work as our generator.

**Testing Classifier Bias Correction Approach.** We demonstrate the utility of correcting for a biased classifier when constructing a dataset of Fair Noise samples. To that end, we compare the distribution of each group when we trust the biased classifier $C_\phi$ (Algorithm 1) to the distribution found when using our proposed biased correction approach (Algorithm 2). We use a linear classifier trained to distinguish images of squares from circles as our biased classifier $C_\phi$. This classifier is implemented using Scikit-learn's LinearSVC classifier [31]. As a ground truth, we utilize a deep convolutional network classifier that achieves approximately perfect accuracy on the task of distinguishing squares from circles. Thus, this network acts as $C'$.

Figure 3 a) shows the distribution of shapes in the Fair Noise Dataset with error correction (Algorithm 2) and without applying our bias correction approach (Algorithm 1). As $|\mathcal{Y}| = 2$ for this experiment, the best case is when the density of each group is $0.5$. Clearly, our bias correction approach yields a dataset that is much closer to being uniform over the semantic attribute. This implies the need to correct for inaccurate classifiers when constructing Fair Noise Datasets.

**Comparative Study on Shapes Dataset.** We next test our proposed Distribution Mapping approach (Section 3.2) against the compared methods. For our approach, we train $M_\omega$ as a CTGAN [43] trained on the corrected Fair Noise Distribution obtained in the previous experiment. As the compared *Latent Editing* method also requires a classifier as a ground truth for training each of its linear models $K_y$, we compare against two versions of the method: 1) *Latent Editing (Biased Classifier)* using the biased $C_\theta$ as a ground truth, and 2) *Latent Editing (Perfect Classifier)* which uses the ground-truth, effectively perfect convolutional network discussed above. Note that from our problem definition, this perfect classifier would usually be unavailable to us.

As Figure 3 b) shows, our approach clearly results in the most uniform distribution out of all compared methods. Additionally, the *Latent Editing* approach performs much worse when using the biased classifier. As expected, *Polarity Antimode Sampling* does indeed increase the frequency of the minority class (*Circles*). Interestingly, *MaGNET* also increases the frequency of the minority class, but is roughly as far from uniform as is the distribution of the original generator; *MaGNET* flips the distribution and under-represents what was previously the majority class.

### 4.3 Age Bias in Face Image Generator

In this next experiment, we evaluate our proposed method's ability to debias a generative model that produces images of people's faces. Specifically, the generative model is a DCGAN [32] that we pretrain on a grayscale version of the UTKFace dataset [44]. For this experiment, the semantic attribute is `Age`. The age of each individual in UTKFace is given as a label. As our approach assumes the semantic attribute $\mathcal{Y}$ is discrete, we bin the ages in increments of 10 years such that $\mathcal{Y} = \{`\leq 9'`, `10\text{-}19'`, \ldots, `90\text{-}99+'\}$. We train a deep convolutional network as the ground truth classifier (used for evaluation), and use a corresponding classifier with a quarter of the feature maps as the biased classifier.

**Evaluating Bias Correction Approach With Large Number of Classes.** We repeat the bias correction experiment we performed on the Shapes data again for the UTKFace Generator. While the previous experiment required only two classes, we now have ten classes (one for each age bin). Despite this increase, Figure 4 a) shows that performing the bias correction we propose in Algorithm 2 yields a much more uniform Fair Noise training dataset than results from trusting the biased classifier.

**Comparative Study on Reducing Age Bias.** We evaluate all methods on the UTKFace Generator, with the goal of making the output images uniform over `Age` (i.e., generate an equal number of images of people belonging to each age group). Figure 4 b) clearly shows that our approach (orange line) produces a much more uniform distribution over ages than the compared method, though it does generate images for the `10-19` bin too infrequently. While approaches such as *Polarity Antimode Sampling* and *MaGNET* are somewhat less biased away from generating images of young people, they fail to generate many samples for older individuals. Note that for the *Latent Editing* approach the we fit a linear regressor (guided by the ground truth) on the latent space rather than a classifier, as age is more naturally a continuous attribute.

### 4.4 Uniformly Sampling Over Race in Progressive GAN

We apply our approach to debias a pretrained Progressive GAN [19]; specifically, the PyTorch [30] version of the 'celebAHQ-256' model [2]. We use the `Race` of the individual in each image as the semantic attribute, and use the MTCNN classifier from the DeepFace [36] package to classify race. Matching the classes available in DeepFace, our semantic attribute space is $\mathcal{Y} = \{`\text{Asian}`, `\text{Indian}`, `\text{White}`, `\text{Middle Eastern}`, `\text{Latino Hispanic}`, `\text{Black}`\}$. The Progressive GAN strongly favors generating white individuals, likely because its training images were of predominantly white celebrities. Here, we consider the classifier to be accurate.

---

[2]see `https://pytorch.org/hub/facebookresearch_pytorch-gan-zoo_pgan/`

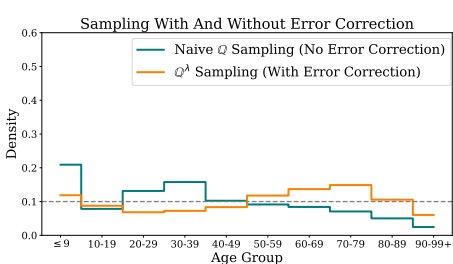

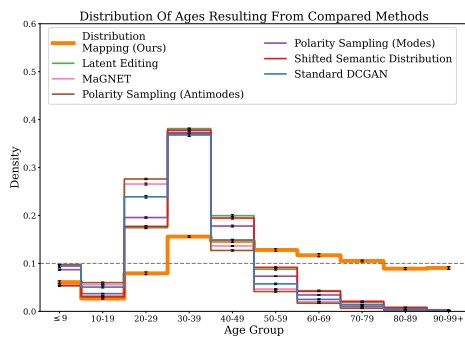

(a) Effect of bias correction; 0.1 (dashed line) is ideal.

(b) Comparison of methods; 0.1 (dashed line) is ideal.

Figure 4: a) Our biased correction approach yields better Fair Noise Distributions; b) comparison of methods on correcting for age bias in the UTKFace DCGAN.

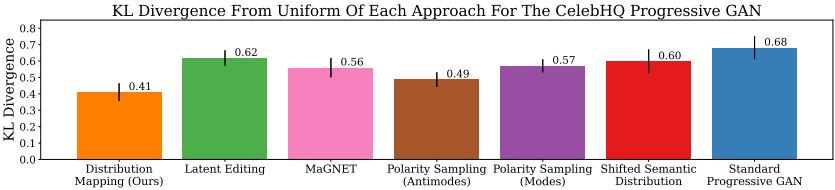

Figure 5: KL Divergence between the distribution over the semantic space for the output of each method (*lower is better*).

**Comparative Study of Debiasing Progressive GAN.** For each method, we report the KL divergence between a uniform distribution and the classifier's output on the samples generated from the method in Figure 5. Note that unlike in the previous experiments, for this metric *lower is better*. Our approach has the lowest KL divergence, indicating it produces a more equal number of images of people from each race than any compared method. We note that despite this we still observe an over-representation of images of white individuals in the samples produced by our (and all other) approaches (see Appendix). This indicates that most regions of the latent space are likely associated with semantic attribute, and unless the distribution mapper very closely matches its Fair Noise training distribution, white persons will still be over-represented. Still, our approach results in a distribution that is most fair out of all compared methods. As we observed previously, sampling antimodes with *Polarity Sampling* produces next-best results; likely because it explicitly draws from low-probability regions of the latent space, which correspond to non-white individuals.

### 4.5 Uniformly Sampling Age in Latent Diffusion Model

Although our focus is primrily on models that map from low dimensional latent space to a higher dimensional feature space, we also evaluate our approach on a latent diffusion model [3] [35] that was trained on the Celeba-HQ dataset [19]. For this experiment we chose Age as the semantic attribute, and debias according to the following age categories: $\{ \leq 29, 30\text{--}49, \geq 50 \}$. We utilize a pretrained ViT age classifier [4] to provide feedback for the semantic attribute. Since our approach requires each latent code to be mapped to a single output image, we utilized the DDIM [41] sampler for the diffusion model to yield a deterministic diffusion process. We used a DDIM diffusion model to train the Mapper network as well.

#### 4.5.1 Comparative Study on Debiasing Latent DDPM.

We report the KL divergence from uniform for the semantic distribution of each compared method in Table 1. Note that we exclude MaGNET and Polarity Sampling from this experiment, due to

---

[3]We utilized this model: https://huggingface.co/CompVis/ldm-celebahq-256

[4]https://huggingface.co/nateraw/vit-age-classifier

| | Distribution Mapping (Ours) | Latent Editing | Vanilla Latent Diffusion |
|---|---|---|---|
| KL Divergence From Uniform Distribution For Semantic Attribute "Age" (Lower is Better) | **0.130** | 0.525 | 0.411 |

Table 1: Comparative study on debiasing the `Age` attribute for a pretrained LDM.

the difficulty of obtaining the Jacobian determinant required by these methods. Results show that our approach yields significantly better scores than the other compared method, indicating that our approach can potentially be useful for debiasing Diffusion models as well.

## 5 Broader Impact And Limitations

The goal of this work, reconditioning generative models to not reproduce the biased distribution of their training set but rather produce one that treats each group equitably, is driven by the desire to mitigate the effects that systemic biases have on generative models increasingly used in real world applications. This has the potential for beneficial societal impact by counteracting the harmful effects of such systemic biases that lead these models to be unfair to underrepresented groups. However, the potential negative impacts of advancing generative modeling should not go unconsidered. Such models have already been used for some harmful applications such as Deep Fakes [28]. It is important to stress that while our approach aims to mitigate the effects of bias in existing models, it is not a fix-all nor an excuse to train models on knowingly biased data. When possible, it is essential to collect fair and equitable training datasets, and to take measures to ensure that the models we train are fair *without* the need for post-hoc corrections.

**Limitations.** Currently, our approach assumes that the semantic space $\mathcal{Y}$ is discrete. We plan to extend this work to handle continuous attributes, such as skin tone, in future work. Additionally, while the conditional distribution $P(\mathbf{x}|\mathbf{y})$ should be roughly equal before and after applying our method if the classifier $C_\phi$ is accurate, in the case of a biased classifier the conditional distribution may change. Correcting for potential distribution shift is likewise future work.

## 6 Conclusion

This is the first method to correct a pretrained biased generative model (i.e., one that strongly favors generating images of white people over all other races) given only the generator and a potentially-biased classifier. We propose a sampling strategy to construct a fair training set using the biased classifier in a way that corrects for its bias, and a Distribution Mapping module that uses this training set to learn how to sample noise instances that produce fair outputs when used as input to the generative model. Notably, we are able to debias the generative model without retraining the model or utilizing any real data. Our results indicate that our approach produces outputs that are much fairer than existing methods. This work may inspire more research on Distribution Mapping techniques to recondition generative models by transforming their standard latent distributions into distributions that yield more favorable behavior.

## 7 Acknowledgments

This material is based on research sponsored by DARPA under agreement number FA8750-18-2-0077. The U.S. Government is authorized to reproduce and distribute reprints for Governmental purposes notwithstanding any copyright notation thereon. The views and conclusions contained herein are those of the authors and should not be interpreted as necessarily representing the official policies or endorsements, either expressed or implied, of DARPA or the U.S. Government.

Results in this paper were obtained in part using a high-performance computing system acquired through NSF MRI grant DMS-1337943 to WPI. We thank the members of WPI's DAISY Research lab and the WPI WASH research group for their insightful feedback and support during the development of this work.

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

# A   Appendix

## A.1   Calculating $E$ for The Generated Distribution

The error rates reported for a classifier $C_\phi$ are typically reported on the distribution on the distribution of fit's training data, $\mathbb{P}_{training}$. However, the distribution $\mathbb{P}_{G_\theta}$ of the generative model $G_\theta$ may differ from the training distribution. Additionally, rather than reporting $P(\mathbf{y}|\hat{\mathbf{y}})$, often times the error rates are given in a confusion matrix $C_{\hat{\mathbf{y}}|\mathbf{y}}$ where $C_{\hat{\mathbf{y}}|\mathbf{y}}[i,j] = P(\hat{\mathbf{y}}|\mathbf{y})$. Thankfully, we can construct the error rate matrix $E$ for the generative distribution $\mathbb{P}_{G_\theta}$ under the simplifying assumption that the difference between $\mathbb{P}_{G_\theta}$ and $\mathbb{P}_{training}$ can be explained as a label shift [10, 40].

By Bayes' Theorem, we know that

$$P(\mathbf{y}|\hat{\mathbf{y}}) = P(\hat{\mathbf{y}}|\mathbf{y})\frac{P(\mathbf{y})}{P(\hat{\mathbf{y}})}.$$

Under the label shift assumption, $P(\hat{\mathbf{y}}|\mathbf{y})$ stays the same between $\mathbb{P}_{training}$ and $\mathbb{P}_{G_\theta}$. Additionally, $P(\mathbf{y})$ can be calculated for $\mathbb{P}_{G_\theta}$ under label shift [10, 40]. Lastly, $P(\hat{\mathbf{y}})$ can be approximated for $\mathbb{P}_{G_\theta}$ by finding the proportion predicted for each class on a large sample from the generative model. Thus, $E$ can be calculated as:

$$E = C_{\hat{\mathbf{y}}|\mathbf{y}}\frac{P_{G_\theta}(\mathbf{y})}{P_{G_\theta}(\hat{\mathbf{y}})}.$$

## A.2   Distribution of Races Generated By Progressive GAN

We show the two best performing methods' distributions on Progressive GAN, along with the distribution of the unmodified ProgressiveGAN, over the `Race` attribute.

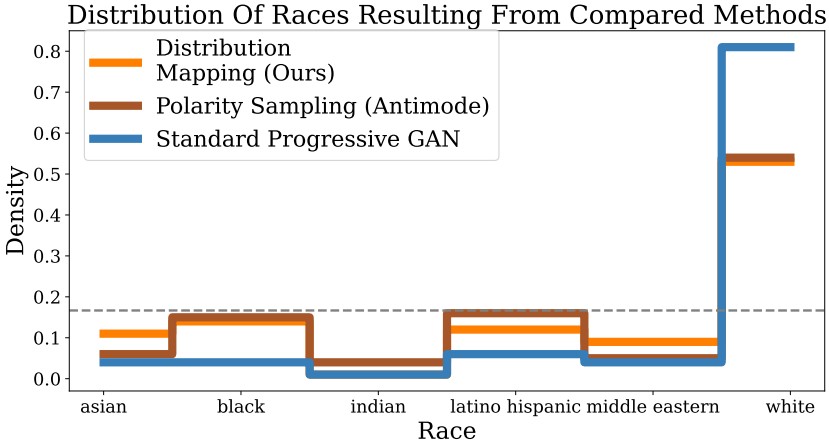

Figure 6: Distribution of our approach, Polarity Antimode Sampling (next best), and the standard generator.

## A.3   Comparison Of Generated Faces From LDM Model

A comparison of a bath of random faces from the original LDM is shown in Figure 7, while Figure 8 shows a batch from the LDM when our Distribution Mapping method is applied.

## A.4   Implementation Details

**Ground Truth Shape Classifier**

```
----------------------------------------------------------------
    Layer (type)              Output Shape
```

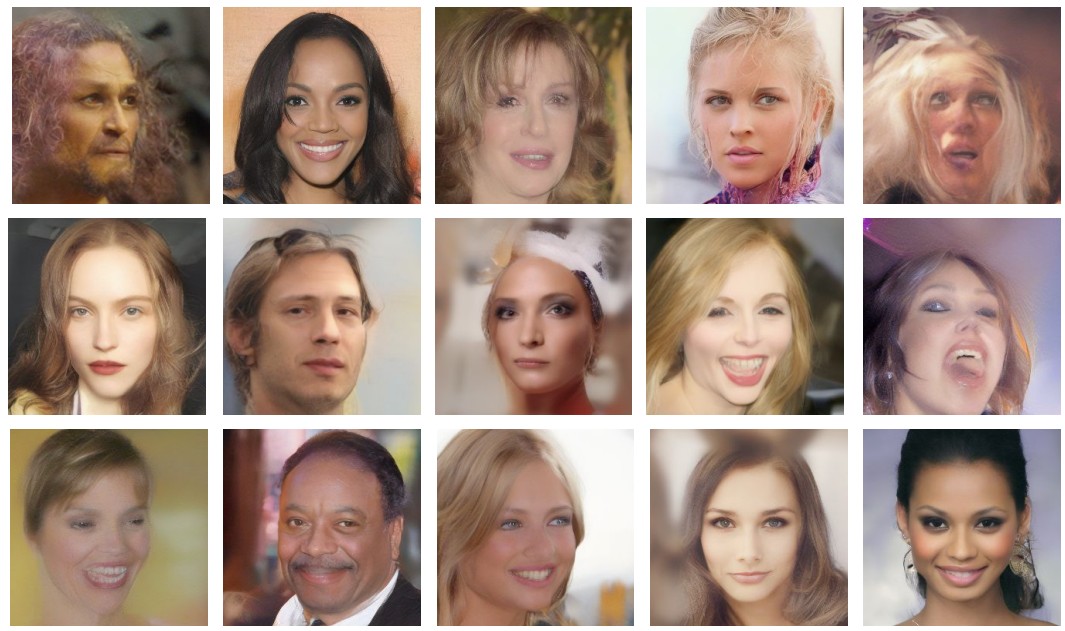

Figure 7: Random sample of generated faces from the original LDM model (no Distribution Mapping or fail sampling applied).

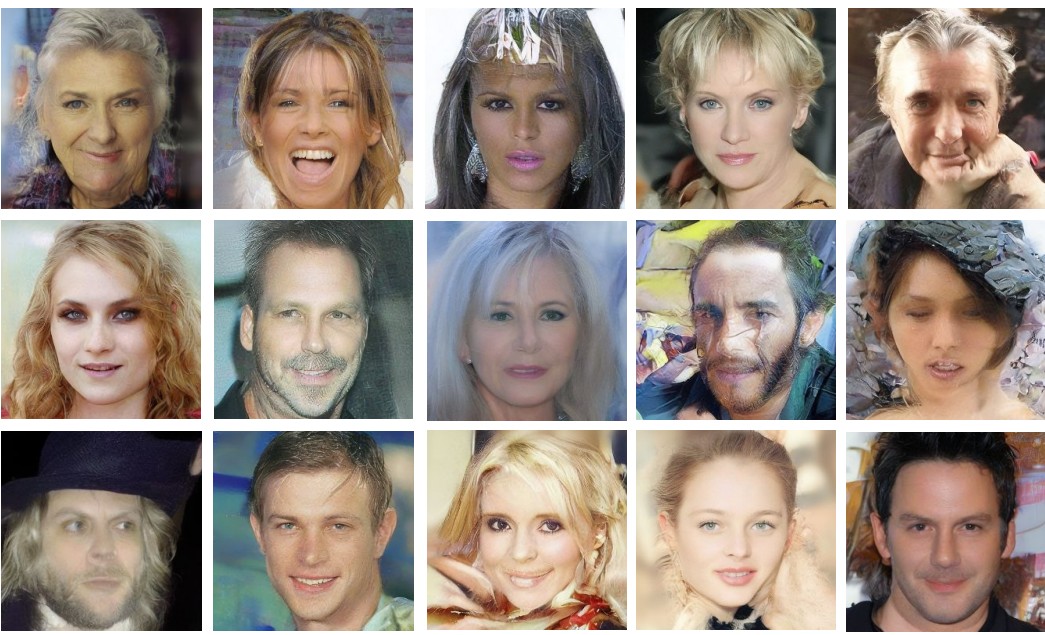

Figure 8: Random sample of generated faces from the LDM model after our Distribution Mapping approach.

```
=================================================================
       Conv2d-1            [-1, 32, 16, 16]
        ReLU-2             [-1, 32, 16, 16]
       Conv2d-3             [-1, 64, 8, 8]
        ReLU-4              [-1, 64, 8, 8]
       Conv2d-5            [-1, 128, 4, 4]
        ReLU-6             [-1, 128, 4, 4]
       Conv2d-7            [-1, 256, 2, 2]
        ReLU-8             [-1, 256, 2, 2]
       Conv2d-9              [-1, 2, 1, 1]
=================================================================
```

**Encoder for Shapes VAE**

```
-----------------------------------------------------------------
        Layer (type)               Output Shape
=================================================================
       Conv2d-1            [-1, 32, 16, 16]
        ReLU-2             [-1, 32, 16, 16]
       Conv2d-3             [-1, 64, 8, 8]
        ReLU-4              [-1, 64, 8, 8]
       Conv2d-5            [-1, 128, 4, 4]
        ReLU-6             [-1, 128, 4, 4]
       Conv2d-7            [-1, 256, 2, 2]
        ReLU-8             [-1, 256, 2, 2]
       Conv2d-9       [-1, code_dim, 1, 1]
=================================================================
```

**Decoder for Shapes VAE**

```
-----------------------------------------------------------------
        Layer (type)               Output Shape
=================================================================
 ConvTranspose2d-1          [-1, 256, 2, 2]
          ReLU-2            [-1, 256, 2, 2]
 ConvTranspose2d-3          [-1, 128, 8, 8]
          ReLU-4            [-1, 128, 8, 8]
 ConvTranspose2d-5         [-1, 64, 16, 16]
          ReLU-6           [-1, 64, 16, 16]
 ConvTranspose2d-7         [-1, 32, 32, 32]
          ReLU-8           [-1, 32, 32, 32]
 ConvTranspose2d-9          [-1, 3, 64, 64]
       Sigmoid-10           [-1, 3, 64, 64]
=================================================================
```

**Biased Age Classifier (Note: Target value was normalized age, made binary after)**

```
-----------------------------------------------------------------
        Layer (type)               Output Shape
=================================================================
       Conv2d-1             [-1, 2, 32, 32]
  BatchNorm2d-2             [-1, 2, 32, 32]
   LeakyReLU-3              [-1, 2, 32, 32]
     Dropout-4              [-1, 2, 32, 32]
       Conv2d-5             [-1, 4, 16, 16]
  BatchNorm2d-6             [-1, 4, 16, 16]
   LeakyReLU-7              [-1, 4, 16, 16]
     Dropout-8              [-1, 4, 16, 16]
       Conv2d-9               [-1, 8, 8, 8]
```

```
        BatchNorm2d-10                   [-1, 8, 8, 8]
         LeakyReLU-11                    [-1, 8, 8, 8]
          Dropout-12                     [-1, 8, 8, 8]
          Flatten-13                       [-1, 512]
           Linear-14                        [-1, 64]
        LeakyReLU-15                        [-1, 64]
           Linear-16                         [-1, 1]
          Sigmoid-17                         [-1, 1]
================================================================
```

**Ground Truth Age Classifier (Note: Target value was normalized age; made binary after)**

```
----------------------------------------------------------------
        Layer (type)                   Output Shape
================================================================
            Conv2d-1                  [-1, 8, 32, 32]
       BatchNorm2d-2                  [-1, 8, 32, 32]
         LeakyReLU-3                  [-1, 8, 32, 32]
           Dropout-4                  [-1, 8, 32, 32]
            Conv2d-5                 [-1, 16, 16, 16]
       BatchNorm2d-6                 [-1, 16, 16, 16]
         LeakyReLU-7                 [-1, 16, 16, 16]
           Dropout-8                 [-1, 16, 16, 16]
            Conv2d-9                  [-1, 32, 8, 8]
       BatchNorm2d-10                 [-1, 32, 8, 8]
         LeakyReLU-11                 [-1, 32, 8, 8]
          Dropout-12                  [-1, 32, 8, 8]
          Flatten-13                      [-1, 2048]
           Linear-14                        [-1, 64]
        LeakyReLU-15                        [-1, 64]
           Linear-16                         [-1, 1]
          Sigmoid-17                         [-1, 1]
================================================================
```

The distribution mapper used default architecture of SDV's CTGAN [5] version 0.6.0, except for in the ProgressiveGAN experiment where `embedding_dim =512`, `generator_dim =(512,512)` were passed as arguments.

For the networks we trained, we utilized the Adam optimizer with learning rate between 0.002 and 0.0001.

The linear classifier utilized Scikit-Learn's LinearSVC (for latent editing) and RidgeClassifier for the biased Shapes classifier.

### A.5 Proof of Lemma 1

*Proof.* First, note that if $1^{|\mathcal{Y}|} \in \text{cone}(E)$, then likewise $\frac{1}{y} 1^{|\mathcal{Y}|} \in \text{cone}(E)$.

Let $\mathbf{z}' \sim \mathbb{P}_{z|C_\phi=i}$; i.e., $z$ is a draw from the distribution of noise such that the classifiers prediction of the generated sample corresponding to $z'$ is group $i$.

Let $(C' \circ G_\theta)_* \mathbb{P}_{z|C_\phi=i}$ be the pushforward distribution of the perfect classifier $C'$'s output when conditioned on the generator's output of draws from $\mathbb{P}_{z|C_\phi=i}$. Then,

$$(C' \circ G_\theta)_* \mathbb{P}_{z|C_\phi=i} = [Pr(\mathbf{y} = 1|C_\theta = i), Pr(\mathbf{y} = 2|C_\theta = i), \dots, Pr(\mathbf{y} = N|C_\theta = i)]$$
$$= E_{:,i}$$

Thus,

---

[5] https://sdv.dev/SDV/user_guides/single_table/ctgan.html# how-to-modify-the-ctgan-hyperparameters

$$\text{cone}(\{(C' \circ G_\theta)_* \mathbb{P}_{z|C_\phi = i}, \ldots, (C' \circ G_\theta)_* \mathbb{P}_{z|C_\phi = |\mathcal{Y}|}\}) = \text{cone}(E)$$

Therefor, following from above,

$$\frac{1}{\mathcal{Y}} 1^{|\mathcal{Y}|} \in \text{cone}(\{(C' \circ G_\theta)_* \mathbb{P}_{z|C_\phi = i}, \ldots, (C' \circ G_\theta)_* \mathbb{P}_{z|C_\phi = |\mathcal{Y}|}\})$$

This means that $\exists \lambda_1, \lambda_2, \ldots, \lambda_{|\mathcal{Y}}$ such that the following holds:

$$\lambda_1 (C' \circ G_\theta)_* \mathbb{P}_{z|C_\phi = i} + \cdots + \lambda_{|\mathcal{Y}|} (C' \circ G_\theta)_* \mathbb{P}_{z|C_\phi = |\mathcal{Y}|} = [\frac{1}{|\mathcal{Y}|}, \ldots, \frac{1}{|\mathcal{Y}|}]$$
$$= \text{Unif}(\mathcal{Y})$$

This is equivalent to saying that:

$$C'(G_\theta(\mathbf{z})) \sim \text{Unif}(\mathcal{Y})$$

for $\mathbf{z} \sim \sum_{i=1}^{|\mathcal{Y}|} \lambda_i \mathbb{P}_{z|C_\phi = i} = \mathbb{Q}^\lambda$. Thus, by definition $\mathbb{Q}^\lambda$ is a Fair Noise Distribution.

$\square$

### A.6 Proof of Lemma 2

*Proof.* Note that the sign of the coefficient of the cross product $E_{:,1} \times E_{:,2}$ is $P(\mathbf{y} = 1|\hat{\mathbf{y}} = 1)P(\mathbf{y} = 2|\hat{\mathbf{y}} = 2) - P(\mathbf{y} = 1|\hat{\mathbf{y}} = 2)P(\mathbf{y} = 2|\hat{\mathbf{y}} = 1)$. Also note that $E_{:,1} \times [0.5, 0.5]$ is $0.5P(\mathbf{y} = 1|\hat{\mathbf{y}} = 1) - 0.5P(\mathbf{y} = 2|\hat{\mathbf{y}} = 1)$.

Additionally,

$$P(\mathbf{y} = 1|\hat{\mathbf{y}} = 1)P(\mathbf{y} = 2|\hat{\mathbf{y}} = 2) > P(\mathbf{y} = 1|\hat{\mathbf{y}} = 1)0.5$$
$$> 0$$

and

$$0 < P(\mathbf{y} = 1|\hat{\mathbf{y}} = 2)P(\mathbf{y} = 2|\hat{\mathbf{y}} = 1)$$
$$< 0.5P(\mathbf{y} = 2|\hat{\mathbf{y}} = 1)$$

Thus, the coefficient of $E_{:,1} \times E_{:,2}$ is greater than $E_{:,1} \times [0.5, 0.5]$, while there signs are equal. This implies that $[0.5, 0.5]$ is in between $E_{:,1}$ and $E_{:,2}$. Thus, $[0.5, 0.5] \in \text{cone}(E)$. The rest of the proof follows directly from Lemma 1. $\square$

### A.7 Proof of Proposition 1

*Proof.* Note that $\mathbb{P}_E^\lambda$ has density $[\sum_i \lambda_i Pr(\mathbf{y} = 1|\hat{\mathbf{y}} = i), \ldots, \sum_i \lambda_i Pr(\mathbf{y} = N|\hat{\mathbf{y}} = i)]$. For ease of notation let us refer to $\sum_i \lambda_i Pr(\mathbf{y} = m|\hat{\mathbf{y}} = i)$ as $r_m^\lambda$.

Then,

$$\text{KL}\{\mathbb{P}_E^\lambda || \text{Unif}(\mathcal{Y})\} = \sum_m r_m^\lambda \log\left(\frac{r_m^\lambda}{u}\right)$$
$$= \sum_m \left(r_m^\lambda \log(r_m^\lambda) - r_m^\lambda \log(\frac{1}{|\mathcal{Y}|})\right)$$
$$= \sum_m r_m^\lambda \log(r_m^\lambda) - \sum_m r_m^\lambda \log(\frac{1}{|\mathcal{Y}|})$$

Note that $\log\left(\frac{1}{N}\right)$ is constant for each term in the second summation. Thus,

$$= \sum_m r_m^\lambda \log(r_m^\lambda) - \log\left(\frac{1}{N}\right) \sum_m r_m^\lambda$$

$$= \sum_m r_m^\lambda \log(r_m^\lambda) - \log\left(\frac{1}{N}\right),$$

As $\log\left(\frac{1}{N}\right)$ does not depend on $r_m^\lambda$,

$$\operatorname*{argmin}_\lambda \mathrm{KL}\{\mathbb{P}_E^\lambda || \mathrm{Unif}(\mathcal{Y})\} = \operatorname*{argmin}_\lambda \sum_m r_m^\lambda \log(r_m^\lambda)$$

$$= \operatorname*{argmin}_\lambda -H(\mathbb{P}_E^\lambda)$$

$$= \operatorname*{argmax}_\lambda H(\mathbb{P}_E^\lambda)$$

$\square$

## A.8  Proof of Proposition 2

*Proof.*

$$(C' \circ G_\theta)_* \mathbb{P}_{z|C_\phi=i} = [Pr(\mathbf{y}=1|C_\theta=i), Pr(\mathbf{y}=2|C_\theta=i), \ldots, Pr(\mathbf{y}=N|C_\theta=i)]$$

$$\implies \sum_i \lambda_i (C' \circ G_\theta)_* \mathbb{P}_{z|C_\phi=i} = [\sum_i \lambda Pr(\mathbf{y}=1|C_\theta=i), \sum_i Pr(\mathbf{y}=2|C_\theta=i),$$

$$\ldots, \sum_i \lambda Pr(\mathbf{y}=N|C_\theta=i)]$$

$$\implies \mathbb{P}_E = \mathbb{Q}^\lambda$$

$\square$

## A.9  Proof of Theorem 1

The first statement follows directly from Proposition 1 and Proposition 2.

If $C_\phi = C'$, then $\{\frac{E_{:,1}}{|E_{:,1}|}, \ldots, \frac{E_{:,|\mathcal{Y}|}}{|E_{:,|\mathcal{Y}|}|}\}$ forms a standard basis of $\mathbb{R}^{|\mathcal{Y}|}$, and therefor $1^{|\mathcal{Y}|}$ is in $\mathrm{cone}(E)$. Thus, $\mathcal{Q}^{\lambda*}$ is a Fair Noise Distribution by Lemma 1.

