# OpenReview forum: "Debiasing Pretrained Generative Models by Uniformly Sampling Semantic Attributes"
_NeurIPS.cc/2023/Conference — NeurIPS 2023 poster_

### Official Review · Reviewer_h8Jf · 2023-07-05

**Soundness:** 3 good
**Presentation:** 4 excellent
**Contribution:** 3 good
**Rating:** 7
**Confidence:** 2

**Summary:**

This paper studies how to learn a fair latent distribution so that a pretrained generator outputs samples that have equal likelihood to fall into different attributes. The paper assumes access to an imperfect attribute classifier. The proposed method learns the fair latent distribution by maximizing the entropy of $P_E^{\lambda}$ and then use a generative model to distill the fair distribution. It is guaranteed that the optimal solution leads to a perfectly fair or minimally unfair distribution. The paper conducts experiments on several image datasets and shows that the proposed method leads to the best fairness in terms of shapes, ages, and races.

**Strengths:**

The paper is extremely well written and organized. It is quite easy to follow and understand the difficulties of the defined task and the motivation behind the solution.

It also tries to solve an important problem in fairness research as modern deep generative models are very powerful but the fairness issues are not well understood.

The proposed method is simple but theoretically very sound. The paper clearly states conditions when the optimal fair distribution may be obtained and when the best possible one is obtained.

The experiments are also extensive. It includes several different sets of attributes and the results look very good to me.

**Weaknesses:**

The experiments do not demonstrate the approximation error or tradeoff of each component of the proposed method. For example, how well does the network learned in section 3.2 approximate $Q^{\lambda}$? How faraway is the learned $P_E^{\lambda}$ from the optimal one? Some visualization or quantitative study for this can help us understand how we may be able to further improve fairness.

Minor points: I also recommend the authors to fix a few typos in the math equations and replace the italic words with \mathrm to make them look nicer.

**Questions:**

Please refer to the weakness comments.

**Limitations:**

The paper discussed limitations in the last section.

---

> ### Author Rebuttal · Authors · 2023-08-10
>
> Thank you for your feedback; and are happy to hear of your favorable view of our work.
>
> **How far off are the learned weights of the mixture of class probabilities from optimal, and how close does the output of the Mapper match the target Minimally-Unfair Noise Distribution?**
>
> We have added another set of experiments to explore your proposed question; please see Table 2 in the rebuttal PDF attached to the general rebuttal post for the new experimental results. In short, the learned weights are very close to optimal.
>
> **Typos and formatting suggestions**
>
> Thank you for the suggestion. We will use \mathrm, and do a thorough pass for typos for the camera ready version.

---

### Official Review · Reviewer_K9u1 · 2023-07-05

**Soundness:** 3 good
**Presentation:** 4 excellent
**Contribution:** 3 good
**Rating:** 6
**Confidence:** 3

**Summary:**

This paper introduces a method for mitigating biases in a pre-trained generator, without the need to retrain the generator or rely on real data. The authors propose the use of a classifier to create a fair distribution of noise vectors, which are then sampled to generate an equal number of instances for each group. Additionally, the paper presents techniques for constructing a fair noise distribution in scenarios where the classifier is either perfect or imperfect.

**Strengths:**

1. The paper's motivation is compelling, considering the widespread adoption of Generative AI across various applications. The problem addressed in the paper is of utmost importance in this context.
2. The authors provide comprehensive results obtained from experiments conducted on three relevant datasets: Shapes, UTKFace, and CelebA-HQ datasets.
3. The paper is well-structured and easy to follow.

**Weaknesses:**

The proposed approach demonstrates a degradation in performance in the case of Progressive GAN applied to the CelebAHQ dataset. The paper includes a discussion that attributes this decline to the observation that a majority of the transformed latent space regions are predominantly associated with the over-represented semantic attribute. This limitation hampers the effectiveness of the proposed approach when dealing with complex datasets. I believe that this problem arises due to the fact that the approach does not involve tweaking the generator function (no retraining involved) but only a transformation of the latent space of the generator.

**Questions:**

The paper lacks a discussion and comparison with relevant baselines, such as [1] and [2], which address the same issue of debiasing generative models without requiring retraining. Given that the authors have included results for Progressive GAN, it would be intriguing to see a comparison between the proposed approach and the aforementioned papers. This would provide valuable insights into the effectiveness of the proposed method in relation to existing techniques.

[1] Karakas, C., Dirik, A., Yalçınkaya, E., & Yanardag, P. (2022). FairStyle: Debiasing StyleGAN2 with Style Channel Manipulations. European Conference on Computer Vision.

[2] Tan, S., Shen, Y., & Zhou, B. (2020). Improving the Fairness of Deep Generative Models without Retraining. arXiv Preprint arXiv:2012. 04842.

**Limitations:**

The authors have addressed the limitations.

---

> ### Author Rebuttal · Authors · 2023-08-10
>
> Thank you for your valuable feedback.
>
> **The paper lacks a discussion and comparison with baselines, such as [1] and [2], which address debiasing generative models without requiring retraining.**
>
> As you have suggested, we have conducted new experiments to compare against the method proposed in [2]; please see Figure 1 in the Rebuttal document attached to the general rebuttal post for results. Our results confirm that our proposed method continues to be the best performer.
>
> As  the method proposed in [1] is designed only for StyleGAN methods - e.g, it requires concepts like style channels and style vectors that are not generally present in other generative models - it is not directly applicable to the models we study.   But, based on your suggestion, we will add a detailed discussion around [1] and its relationship to our work into our related work section.

---

> > ### Comment · Reviewer_K9u1 · 2023-08-16
> >
> > Thank you for your response.
> >
> > Also, thank you for conducting the experiments that compare against the approach proposed in [2]. The result in the document clearly shows that the proposed approach performs better compared to [2].

---

> > > ### Comment · Reviewer_K9u1 · 2023-08-18
> > >
> > > Since my concerns have been addressed, I have increased my score from 5 to 6.

---

> > > ### Author Response · Authors · 2023-08-21
> > > **Thank you for increasing your score**
> > >
> > > Thank you again for your feedback alerting us to these related works, and for raising your score.

---

### Official Review · Reviewer_CzLf · 2023-07-06

**Soundness:** 2 fair
**Presentation:** 3 good
**Contribution:** 3 good
**Rating:** 5
**Confidence:** 4

**Summary:**

The paper proposes a method for training a Distribution Mapper that can generate a noise distribution capable of creating Fair Samples using an Imperfect Classifier. They conducted experiments on toy examples and several Face Generation models, demonstrating quantitatively superior performance compared to competitors.

**Strengths:**

The paper proposes an effective approach to achieve debiasing without the need for additional training of generative models or classifiers by ingeniously devising a method to train $M$.

**Weaknesses:**

Although it is mentioned in Line 114 that $M$ could be GAN, VAE, DDPM, normalizing flow, etc., there are no additional experiments conducted to explore this further. There is a lack of experimentation to examine how the noise shifted by $M$ affects image quality. It would be beneficial to report FID scores for CelebA as well. While $G$ is defined as a generative model, there are only experiments conducted with GAN. Particularly for Diffusion models, it is doubtful whether $M$ can be effectively trained.

**Questions:**

Does the verification of debiasing using metrics such as CLIP Score yield similar results?

**Limitations:**

Please refer to the Weakness part. I have a positive view of the content and approach presented in this paper. It would be beneficial to explore different models for M, not just limited to GANs, and investigate their performance. Additionally, conducting experiments on more recent models trained on datasets like CelebA and FFHQ would be valuable. It would also be interesting to observe the tendencies of commonly used models such as StyleGAN2, DDPM, LDM, VAE, and others. Including these aspects would enhance the quality of the paper and make it even more comprehensive.

---

> ### Author Rebuttal · Authors · 2023-08-10
>
> Thank you for the time you put into reviewing our work, and for the constructive feedback you provided.
>
> **It would be beneficial to report FID scores for CelebA as well.**
>
> We computed the FID score for the Progressive GAN before and after applying our method, and found that the increase in FID was not extreme; it increased around 5 points. Since the distribution of CelebA is biased towards images of white people, it is not surprising that a debiased distribution would have an increased FID score.
>
> **There are only experiments conducted with GAN.**
>
> There is a misconception that we experimented with GANs only –  which must mean that we had not adequately highlighted that the generative model we worked with in the experiments in Section 4.2 was a VAE model. We will more explicitly state the generative models studied in the camera ready version.
>
> **Could this work for Diffusion?**
>
> Our target when designing our approach was focused on generative models with a latent space that is of lower dimensionality than the data space – which matches GANs that indeed were our key target.  With Diffusion models having a latent space in dimensionality equal to that of the data space, we agree that training the mapper will be more difficult.
>
> However, it is still conceptually possible. As a proof of concept, we have added an experiment on a latent diffusion model [1], where the diffusion model acts on 3x64x64 latent space - which is much larger than those of the GANs and VAE we have studied in our original manuscript. Our results show that our method indeed yields a distribution over ages that is closer to uniform than those produced by any of the other compared approaches; please see Table 1 in the Rebuttal pdf attached to the general Rebuttal post.
>
> **Conducting an experiment on more recent models would be valuable.**
>
> Based on your suggestion, we conducted a new experiment described above on the latent diffusion model,  a recent model. We will add these additional results into the final manuscript.
>
> **Additional experiments on using methods other than GANs for the mapper M would be valuable.**
>
> To show that the mapper could be some other generative model besides a GAN, we use a DDIM diffusion model as the mapper for the above latent diffusion experiment. We use a mapper with less than half as many parameters as the U-net of the pretrained latent diffusion model - and thus training the mapper is much cheaper than retraining the generative model.
>
> [1] Rombach et al., “High-Resolution Image Synthesis with Latent Diffusion Models” CVPR 2022.

---

> > ### Comment · Reviewer_CzLf · 2023-08-14
> >
> > I thank the authors for their answers. Also, I'm sorry that I missed the experiment on VAEs.
> >
> > My questions were addressed adequately but few questions remain.
> >
> > As reviewer yDHD mentioned, if the authors modify the initial noise x_T~N(0,I), it is very hard to maintain the quality of outputs. In my acknowledgment, for now, nobody could edit the image in Gaussian space, x_T. Could you give me more information about the experiment on LDM?

---

> > > ### Author Response · Authors · 2023-08-16
> > > **Additional Clarifications**
> > >
> > > We are happy to hear that most of your questions have been answered sufficiently.
> > >
> > > Indeed, editing in the Gaussian space X_T is a difficult problem which requires more research and may not be feasible at this time with non-deterministic samplers like in a standard DDPM. We note though this may be easier when using diffusion models with **deterministic samplers** such as DDIM, which indeed we utilized. Namely,  the authors of the original DDIM paper report the ability to perform simplistic semantically meaningful latent edits in the form of interpolation [1]. See Table 6 in [1], and the corresponding text: “[We] show that simple interpolations in $x_T$ can lead to semantically meaningful interpolations between two samples”. However, this is indeed not exactly equivalent to the robust latent editing that has been performed using GANs.
> > >
> > > We thus agree that unlike GANs, there are often not clear directions in the X_T space that correspond to particular semantic attributes. In fact, this may likely be the reason why the Latent Editing method performed worse than the baseline in our LDM experiment.
> > >
> > > However, **our approach is not really conducting edits in the latent space**; e.g., we are not using a latent space traversal to add or remove attributes from a generated image. Instead, we are sampling certain regions of the latent space with more or less frequency. This will make some attributes more common on average, but will not allow for editing of specific instances.
> > >
> > > As an example, consider a case where we have a frozen diffusion model that generates pictures of cats and dogs, with more dog images than cat images on average. Let z_1 be a latent code that happens to map to an image of a dog, and z_2 be a latent code that happens to map to an image of a cat. Let us also assume that we have a deterministic sampler for the LDM - in our case, a DDIM sampler - so that z_1 and z_2 will always produce the dog and cat picture respectively.
> > >
> > > Let’s say we have a trivial  “mapper” that learns to yield only z_1 and z_2 with equal probability. If we use this mapper to produce a dataset of noise (e.g. multiple copies of z_1 and z_2), and then pass this to the (deterministic) LDM model, we would produce a dataset of an equal number of dog and cat images (though the dog and cat images would be duplicated). Thus, we changed the distribution of X_T (in this case, it would be a mixture of two dirac delta distributions centered on z_1 and z_2) into one that produces an equal number of valid dog and cat images. However, no “editing” of any image or latent code would have to take place here.
> > >
> > > Of course, in our case we do not use this trivial mapper; we instead have our mapper learn to match a range of samples so we have a variety of output images, not duplicates, but the core principle remains.
> > >
> > > We list the experimental details below. We will add the code for this experiment to the supplemental material for the camera-ready version as well. Currently, we cannot edit the supplemental material.
> > >
> > > —----
> > >
> > > Diffusion model: “ldm-celebahq-256” from CompVis, available on Hugging Face
> > >
> > > Classifier: “vit-age-classifier” from nateraw on Hugging Face
> > >
> > > Mapper: The default U-Net used in the “Train a diffusion model” Hugging Face tutorial, with image size set to 64
> > >
> > > Fair Noise Dataset used to train mapper: 20,000 samples from each of the following bins:
> > > * Predicted Age < 29
> > > * Predicted age 30 - 49
> > > * Predicted age > 50
> > >
> > > Task: Generate samples uniform over the three bins defined above
> > >
> > > [1] Song et al.  "Denoising diffusion implicit models." ICLR 2021.

---

> > > > ### Comment · Reviewer_CzLf · 2023-08-16
> > > >
> > > > I thank the authors for their answers. I have increased my score; 4->5.
> > > >
> > > > I re-read the corrections and some proofs in Appendix, and I suggest a minor thing; A.6 and A.7 are hard to read. Could you re-organize it?

---

> > > > > ### Author Response · Authors · 2023-08-21
> > > > > **Thank you for increasing your score**
> > > > >
> > > > > Thank you for your feedback, snd we are very grateful that you raised your score.
> > > > >
> > > > > We are certainly happy to edit the appendix for our camera ready submission

---

### Official Review · Reviewer_yDHD · 2023-07-26

**Soundness:** 3 good
**Presentation:** 3 good
**Contribution:** 3 good
**Rating:** 7
**Confidence:** 4

**Summary:**

This paper proposes an approach to mitigate biases in pre-trained generative models. Specifically, the authors aim to generate a semantically fair distribution over the attributes. Authors propose to train a distribution mapper network that effectively learns to generate a fair noise distribution, which, in turn, is used to generate samples that adhere to fairness principles such that the distribution of the generated samples per-group is closer to the uniform distribution.
Authors theoretically consider the case where the classifier to train the mapper network is imperfect and propose an approach to handle this case. Furthermore, they study this approach on three tasks, Shapes Dataset, Age Bias and race bias and compare with existing debiasing methods.


-----------------------------------------------------------------------------------------------------
Following Neurips guidelines: I acknowledge that I have read the authors response, engaged in the discussions and read other reviewers comments.  Based on the authors rebuttal, I have updated my score accordingly and increased it from 6 to 7.

**Strengths:**

- The paper addresses an important problem, debiasing generative models output.
- The paper proposes an approach that does not require training or fine-tuning a generative model which is computationally efficient.
- The paper proposes to use a pre-trained classifier that is able to classify groups and they also propose an algorithm that handles imperfect classifiers.
- In the experiments, the paper compares with several other existing methods for debiasing the generative models and in all the reported cases, the method out-performs existing methods.

**Weaknesses:**

- The authors frame this work around the general generative models, however, in the experiments, only GANs are studied. I believe the study of diffusion models or other types of generative models is important. Specifically, diffusion models generate samples stochastically, and changing the initial noise distribution may not be as trivial as GANs. Furthermore, diffusion models have a strict assumption on the starting noise to be from the uniform distribution. Thus, changing the initial noise distribution is not trivial. I believe the paper should be framed around GANs since it is the only model that is studied empirically and is trivial to extend given the setup.
- The authors mention that pre-trained classifiers that classify specific groups/attributes is easily accessible. However, in many cases that may not be the case (likely the issue with the experiment on progressive GANs).
- The experiments are minimal and limited to GANs. It would have been intersting to see experiments related to other types of generative models or extend other types of more large-scale GANs or larger scale datasets.

**Questions:**

- Since the work is framed in a way that supports general generative models, how do you propose to use this approach for VAEs or diffusion models?
- How do you handle the stochasticity that comes from the noise in the GPU training? Do you need the training process of the GAN to be deterministic?

**Limitations:**

authors have already addressed the limitations.

---

> ### Author Rebuttal · Authors · 2023-08-10
>
> We thank you for your thoughtful feedback.
>
> **“How do you handle stochasticity that comes from the noise during training? Do you need the training process of the GAN to be deterministic?”**
>
> We do not require the training of the generative model to be deterministic. We only require that, once trained, the generative model produces a very similar output when conditioned on the same latent code instance. This behavior is satisfied by standard generative models like GANs, VAEs, normalizing flows, and diffusion models if a deterministic reverse process sampler is used.
>
> **“Can this approach work on Diffusion models?”**
>
> We note that indeed the focus of our research when formalizing our approach has been on generative models such as GAN-type architectures where the dimensionality of the latent space is typically much lower than the data space. This is not the case for Diffusion models. However, conceptually, our approach could still be applied to Diffusion models that are paired with a deterministic sampler (such as DDIM). As a proof of concept, we have conducted a new experiment debiasing a Latent Diffusion Model [1] that was trained on Celeba-HQ-256. In this new experiment, we make the “age” attribute of the generated faces more uniform. Our method succeeds to generate faces with a more fair distribution over ages. Please see Table 1 in the Rebuttal PDF attached to the general rebuttal post.
>
> **“Don’t Diffusion models require a specific noise distribution?”**
>
> While diffusion models typically result in a specific distribution for the forward process during training (e.g., Gaussian noise), during inference when performing the reverse process a modified distribution can be sampled from.
>
> Consider the case where we draw a sample $z$ from the normal noise distribution, and use this to generate an output image $x$ using a DDIM sampler for a pretrained frozen Diffusion model. Let us then define a dirac delta distribution on $z$, and then sample from this delta distribution instead of from the original noise distribution. Each of these samples, once run through the diffusion model, will likewise generate the same $x$. Thus, even with a modified noise distribution, the output should still be in the target manifold - though the distribution of the outputs will change. However, changing the output distribution is exactly what we are trying to achieve.
>
> **“Why were experiments only performed on GANs?”**
>
> We believe that there is a misunderstanding here. Namely,  the generative model we evaluate in Section 4.2 is a Variational Autoencoder (see line 247 in our paper). As this misconception was shared by more than one reviewer, we now realize that this point of our work with the VAE model was not highlighted properly in our original manuscript. In the camera ready version, we will thus include a paragraph in the experimental setup (Section 4.1) specifically laying out which generative models we evaluate. Additionally, as mentioned earlier in the rebuttal, we have now also conducted a new experiment on a Diffusion model.
>
> **“How realistic is it to assume that an attribute classifier is available?”**
>
> While there may be some scenarios where publicly available classifiers do not exist for a specific attribute of interest to a given application, we have found that there are a good number of scenarios where there are such classifiers. Thus, while our method will not apply to every scenario, there are many cases where it will. For instance, there are a good number of face attribute classifiers available that support the debiasing images of people, as discussed in our manuscript.
>
> Additionally, as training a classifier is typically easier than training a generative model, one worst-case option would be to train a new classifier on the attribute of interest and use that to debias the pretrained generative model.
>
> **“Since the work is framed in a way that supports general generative models, how do you propose to use this approach for VAEs or diffusion models?”**
>
> We agree that indeed our primary focus was on solving this problem for GAN-like models that map from a low-dimensional latent space to a high-dimensional data space. We will rephrase our claim about generality to indicate that while the approach may be applicable to generative models that have higher dimensional noise distributions, we  are most focused on this type of model that takes in lower dimensional noise.
>
> However, given a latent code, the decoder of a VAE works analogously to the generator of a GAN. It is for this reason that we had experimented with VAE models in this manuscript (Section 4.2),  as explained above.
>
> For a diffusion model, we would transform the initial latent distribution (i.e., the distribution of X_T), and then use a deterministic sampler such as DDIM for the reverse process.
>
> [1] Rombach et al., “High-Resolution Image Synthesis with Latent Diffusion Models” CVPR 2022.

---

> > ### Comment · Reviewer_yDHD · 2023-08-14
> >
> > I would like to thank the authors for carefully answering my questions and concerns. I think most of your reply could be included in the paper as a discussion, specially the one around stochasticity.
> >
> > Also, I'm sorry that I had missed your original experiment on VAEs, thank you for the clarification. I hope that you highlight this experiment in your text.
> >
> > Regarding your new experiment on diffusion models, you mentioned that you have used DDIM, in the DDIM paper (https://arxiv.org/pdf/2010.02502.pdf), Eq. 12, there are multiple noises, 1) x_T which is sampled from N(0, I), and also \epsilon_t, which is also sampled from N(0, I). Are you modifying x_T or \epsilon_t? Because comparing your approach when applied to GANs, modifying the starting noise (x_T) would make more sense. But from your response, I believe you are modifying \epsilon_t? Could you clarify? Are you learning a fair noise distribution over every time step t?
> >
> > I'm also concerned about the quality of the generated samples (also mentioned by reviewer CzLf). I believe there must be a trad-off between quality and diversity of the generated samples. I saw that you mentioned that the FID is increased around 5 points, if that's the case, I believe this is a major degradation of sample quality. It would have been nice to show the trade-off and also see the samples for the latent diffusion model.
> >
> > All in all, given above, I believe this work is valuable and intersting for the Neurips community and I vote for acceptance.

---

> > > ### Author Response · Authors · 2023-08-16
> > > **Thank you for recommending acceptance**
> > >
> > > We are very grateful to you for recommending acceptance. As you suggest, we would be happy to add the details of our response back into the discussion section of the camera-ready paper.
> > >
> > > Regarding your question about DDIM, we confirm that indeed we modify x_T. However, we do not learn a new distribution for each step. Since we use a deterministic sampler (DDIM), the distribution of the following steps (e.g. x_T-1, …, X_0) will be completely determined from the distribution at step T. In Equation 12, which you reference, $\sigma_t$ would be set to 0. Quoting from the DDIM paper:
> > >
> > > “[when \sigma_t is set to 0, ] in the generative process, the coefficient before the random
> > > noise t becomes zero. The resulting model becomes an implicit probabilistic model […] where samples are generated from latent variables with a fixed procedure (from xT to x0). We name this the denoising diffusion implicit model (DDIM)”
> > >
> > > Thus, as you can see, there is no randomness beyond the initial random noise from the draw from X_T, as each $\epsilon_t$ would be multiplied by 0.
> > >
> > > Regarding sample quality: We do not believe the open review system supports for us to upload another pdf document at this time. For this reason, we cannot provide you with examples of our generated images currently. However, using a visual inspection, we can confirm that while there are some visual abnormalities in our outputs, the generated images do not look significantly far off in quality from the average outputs of the pretrained models. We will add visual examples of the output of our method, and compared methods, into the camera ready document.

---

> > > > ### Comment · Reviewer_yDHD · 2023-08-16
> > > >
> > > > This is a very interesting observation that you've had with DDIM. Thank you for your reply. I have increased my score.

---

> > > > > ### Author Response · Authors · 2023-08-21
> > > > > **Thank you for increasing your score**
> > > > >
> > > > > Thank you for the time you spent discussing our work with us, and for raising your score. It is very appreciated.

---

### Author Rebuttal · Authors · 2023-08-10

## General Rebuttal

We thank the reviewers for the time they spent providing valuable feedback on our work. We have added additional experiments (see attached PDF) based on the reviewers’ suggestions.

In short, we:

-  Performed a new experiment, where we apply our approach to a Latent Diffusion Model (Table 1 in attached PDF), as suggested by Reviewer yDHD and Reviewer CzLf.

- We add in a comparison against another related work (Tan et al.) based on the feedback from Reviewer K9u1 (Figure 1).

- We report metrics on how well our method matches its target distribution (Table 2) as requested by Reviewer h8Jf.

Additionally, we want to clarify a common misconception among reviewers: namely, that we only performed experiments on GAN models in the original manuscript. We actually did apply our method to a VAE in the original manuscript (see section 4.2, line 247). We realize that we did not adequately highlight this in our original text; we will explicitly list each generative model, including the new latent diffusion model we added for this rebuttal, in the camera ready version.

---

### Decision · Program_Chairs · 2023-09-21

**Decision:**

Accept (poster)

**Comment:**

This paper was reviewed by four knowledgeable referees. The reviewers raised concerns w.r.t. the experiments presented in the paper, which initially appeared limited (yDHD, CzLf, K9u1); the requirement to have access to a pretrained classifier that classifies groups/attributes (yDHD); and the effect of shifting the noise on the resulting image quality (CzLf). The reviewers also raised questions w.r.t. the trade off of each component (h8Jf). The rebuttal adequately addressed the reviewers' concerns by highlighting results of both GANs and VAEs in the original paper, including additional baselines, and presenting results on diffusion models. After rebuttal and discussion, the reviewers unanimously lean towards acceptance. The AC finds this an interesting and timely contribution and agrees with the reviewers' assessment. Therefore, the AC recommends to accept.